# Receptors and Signaling Pathways Controlling Beta-Cell Function and Survival as Targets for Anti-Diabetic Therapeutic Strategies

**DOI:** 10.3390/cells13151244

**Published:** 2024-07-24

**Authors:** Stéphane Dalle, Amar Abderrahmani

**Affiliations:** 1Institut de Génomique Fonctionnelle, Université de Montpellier, Centre National de la Recherche Scientifique (CNRS), Institut National de la Santé et de la Recherche Médicale (INSERM), 34094 Montpellier, France; 2Université Lille, Centre National de la Recherche Scientifique (CNRS), Centrale Lille, Université Polytechnique Hauts-de-France, UMR 8520, IEMN, F59000 Lille, France

**Keywords:** diabetes, pancreatic beta-cell, insulin secretion, apoptosis, receptors, signaling pathways, therapeutic strategies

## Abstract

Preserving the function and survival of pancreatic beta-cells, in order to achieve long-term glycemic control and prevent complications, is an essential feature for an innovative drug to have clinical value in the treatment of diabetes. Innovative research is developing therapeutic strategies to prevent pathogenic mechanisms and protect beta-cells from the deleterious effects of inflammation and/or chronic hyperglycemia over time. A better understanding of receptors and signaling pathways, and of how they interact with each other in beta-cells, remains crucial and is a prerequisite for any strategy to develop therapeutic tools aimed at modulating beta-cell function and/or mass. Here, we present a comprehensive review of our knowledge on membrane and intracellular receptors and signaling pathways as targets of interest to protect beta-cells from dysfunction and apoptotic death, which opens or could open the way to the development of innovative therapies for diabetes.

## 1. Introduction

Diabetes is a chronic disease that represents a major economic and clinical burden for the healthcare systems of most countries in the world. In recent years, the International Diabetes Federation has drawn an alarming picture of the rising incidence of diabetes worldwide, which is reaching pandemic proportions [1]. The diabetes pandemic is linked to aging populations and obesity resulting from lifestyle changes (reduced physical activity, diets rich in saturated fats and simple sugars) (for reviews, see [1,2,3,4,5,6,7,8]). There are three main types of diabetes: type 1 diabetes (T1D), type 2 diabetes (T2D), which accounts for over 90% of all diabetes cases, and gestational diabetes (for reviews, see [1,2,3,4,5,6,7,8]). Currently, the prevalence of T1D and T2D among people aged 20 to 79 reaches 9% of the world population, with 463 million people affected [1]. It is expected that the prevalence will worsen in future, to 578 million people (10% of the population) in 2030 and 700 million (11% of the population) in 2045 [1].

In T1D and T2D, hyperglycemia occurs because plasma insulin levels cannot meet the organism’s needs [2,3,4,5,6,7,8]. In mammals, insulin is only produced by the beta-cells of the islets of Langerhans, also called the endocrine pancreas [2,3,4,5,6,7,8]. These islets represent ~2% of the total volume of the pancreas and constitute the endocrine organ. The islets are small, compact, and spherical cell clusters [9,10,11]. The population of islets within a human pancreas is heterogeneous in size. Beta-cells represent the majority of the cells within the islets (~50–55% of cells in an adult human islet) [9,10,11]. Besides beta-cells, other cell types are found here, including the alpha-cells synthesizing and secreting glucagon (~35–40% of cells), the delta-cells synthesizing and secreting somatostatin (~2–5% of cells), and small numbers of cells such as the pancreatic polypeptide, ghrelin, and YY peptide cells [9,10,11]. In humans, the islets are organized around a central body of beta-cells surrounded by a crown of non-beta-cells. The non-beta-cells (especially alpha-cells) are attached to the blood vessels located on the periphery of the islets, but these cells also enter the center of the islet. Up to 70% of beta-cells are directly in contact with non-beta-cells [9,10,11]. This islet architecture facilitates functional communication and exchanges between the different cell types. The secretion of one cell type can regulate the function of another cell type, and the membrane functional junctions between homologous and heterologous endocrine cells are well described [9,10,11].

Insulin secretion is triggered by circulating concentrations of glucose [12,13,14,15,16]. Glucose enters the beta-cells through glucose transporters (GLUT1/2) in proportion to its circulating concentration. Within the beta-cell, glucose is phosphorylated by glucose kinase (GK, also known as hexokinase type IV) to glucose-6-phosphate and metabolized through glycolysis in the cytosol and the KREBS cycle (tricarboxylic acid cyclic (TCA)) in the mitochondria to produce ATP, which closes the Kir6.2 permeability pores of the K^+^/ATP channels [12,13,14,15,16]. This prevents the release of K^+^ from the beta-cell, causing localized membrane depolarization that opens voltage-dependent calcium channels (VDCCs) (Figure 1) [12,13,14,15,16]. The influx of calcium increases the cytosolic concentration of calcium ([Ca^2+^]_c_), which triggers insulin secretion through a soluble N-ethylmaleimide-sensitive factor attachment protein receptor (SNAREs)-mediated fusion of a readily releasable pool of insulin-containing vesicles within the plasma membrane (Figure 1) [14,15]. This triggering mechanism is responsible for the first phase of insulin secretion (or the early peak of insulin), which occurs during the first 5–10 min. The second phase, also known as the amplifying pathway, is late, more sustained, and achieved over a period of 30–60 min (for reviews, see [12,13,14,15,16]). The second phase of glucose-induced insulin secretion relies on K^+^/ATP channel-independent mechanisms involving several metabolites including TCA intermediates (NADPH and NADH), associated products (glutamate, malonyl-CoA), phospholipase C/protein kinase C signaling, and an increase in cAMP levels, which together enhance [Ca^2+^]_c_ and insulin granule exocytosis. This second phase is maintained until stimulation stops (for reviews, see [12,13,14,15,16]). Although all beta-cells are able to secrete insulin in response to glucose in a biphasic manner, glucose competence seems to differ between four beta-cell subtypes [17]. The beta1- and beta2-cell subtypes expressing the lowest levels of ST8SIA1 have higher glucose competence than the beta-3 and beta4-cells harboring the highest levels of ST8SIA1 [17].

In vivo, the insulin secretion triggered by glucose is regulated by hormones, growth factors, and neurotransmitters which act through the activation of membrane or intracellular receptors, leading to the stimulation and/or inhibition of signaling pathways [18,19,20]. The multifactorial modulation of beta-cell function requires the rapid integration of a set of information that generates complex and interconnected intracellular signaling pathways. It is acceptable to classify agents that modulate insulin secretion into three categories: initiators that are able to trigger insulin secretion such as glucose; amplifiers, or potentiators, which do not have effects on their own but increase insulin secretion in the presence of an initiator, especially glucose; and attenuators that inhibit the insulin secretion induced by glucose [12,18,19,20]. As examples, incretin hormones such as glucagon like peptide-1 (GLP-1) or gastrointestinal peptide (GIP), produced and released by the intestine following food ingestion, are powerful potentiators of the insulin secretion triggered by glucose [21,22,23,24]. Somatostatin released by delta-cells in the islet inhibits the release of insulin by beta-cells (and also glucagon by alpha-cells) [25,26]. The activation of membrane and/or intracellular receptors and signaling pathways influences insulin secretion mechanisms and regulates the molecular programs controlling the intrinsic properties of beta-cells, such as the maintenance of a “competent” phenotype for the physiological response to blood glucose variations, survival, and proliferation [18,19,20,21,22,23,24,25,26].

Beta-cells play a central role in diabetes’ etiology [2,3,4,5,6,7,8]. In the case of T1D, the autoimmune destruction of beta-cells and reduced beta-cell mass are the main features of the disease [2,3,4]. The destruction of beta-cells is due to inflammatory reactions such as the infiltration of mononuclear cells into the islets, resulting in elevated concentrations of pro-inflammatory cytokines and chemokines [2,3,4]. T1D is considered a heterogeneous disease. In the context of T1D, endotypes have been proposed on the basis of various factors such as age, genetic predisposition, environmental interactions, the level of immune-induced beta-cell destruction, clinical features, and response to therapies [3,4]. For example, a younger age is associated with a higher risk and rate of progression through the stages of the disease and high levels of islet-infiltrating immune cells and beta-cell destruction (T1D endotype 1). Individuals diagnosed later have fewer immune cells infiltrating their islets and residual islets containing insulin despite little or no insulin secretion (T1D endotype 2) [4]. T2D is a complex heterogeneous syndrome of polygenic origin that associates two types of interdependent abnormalities: (1) decreased tissue sensitivity to the effects of insulin (insulin resistance) in skeletal muscles and adipose tissue and (2) insulin secretion dysfunction in response to glucose in beta-cells [5,6,7,8]. The progressive functional exhaustion of beta-cells to compensate for insulin resistance is a crucial event in the emergence of T2D [5,6,7,8]. The defective glucose sensing of beta-cells and loss of the first phase of insulin secretion are hallmarks of T2D [5,6,7,8]. The United Kingdom Prospective Diabetes Study (UKPDS 16) showed that beta-cell function is reduced by ~50% when diabetes is diagnosed [27]. Post-mortem studies have reported a decrease in beta-cell population and mass in patients with T2D [28,29]. The dysfunction and death by apoptosis of beta-cells in T2D are known to be caused by chronic hyperglycemia (glucotoxicity), certain high-concentration fatty acids (lipotoxicity), and islet amyloid polypeptide deposits (for reviews, see [30,31,32,33,34,35,36]). Chronic hyperglycemia promotes a vicious circle leading to the gradual deterioration of beta-cell function and mass and strengthens insulin resistance [30,31,32,33,34,35,36]. Chronic, systemic, and low-grade inflammation was also proposed to induce beta-cell dysfunction and death and T2D [37,38]. The dynamic balance of functional beta-cell mass is a key element for the long-term control of glycemia. This reinforces the importance of studies on the restoration and/or maintenance of a functional beta-cell mass in diabetic patients in the research for new therapeutic options.

In the next few years, diabetes will unfortunately continue to affect a large and growing proportion of the world’s population. The progress observed in the prevention of overweight and obesity, and the encouragement of more physical activity, is promising but still insufficient. Technological solutions are focused on the development of an artificial pancreas, while biological solutions aim to transplant a pancreas, islets of Langerhans, or other cell types that can be engineered and used safely to replace beta-cells. Intensive research into reducing symptoms such as hyperglycemia has led to the marketing of an increasing number of hypoglycemic agents. However, the effectiveness of the current available treatments decreases over time, and their administration presents risks of effects degrading the quality of life of the patient. In addition, none of the anti-diabetic drugs currently in clinical use promote the maintenance of a functional beta-cell mass over time, revealing an unmet medical need. Therefore, the search for drugs and therapeutic strategies specifically protecting beta-cells from the deleterious effects of inflammation and/or chronic hyperglycemia over time is a priority.

A better understanding of receptors and signaling pathways, and of how they interact with each other in beta-cells, remains crucial and is a prerequisite for any strategy to develop therapeutic tools aimed at modulating beta-cell function and/or mass. Here, we comprehensively review our knowledge of membrane and intracellular receptors and signaling pathways as targets of pharmacological interest to protect beta-cells against dysfunction and apoptotic death, which pave or could pave the way for the development of innovative therapies for diabetes.

## 2. Membrane Receptors

### 2.1. G Protein-Coupled Receptors Regulating Insulin Secretion and Beta-Cell Survival

A large variety of ligands activate G protein-coupled receptors (GPCRs) including photons, peptides, and proteins (for reviews, see [39,40,41,42,43]). There are more than 800 GPCRs involved in various physiological processes such as vision, neuronal functions, cardiovascular functions, endocrine functions, or reproduction. This diversity of functions makes this family of receptors a prime target for the development of therapeutic strategies. More than 35% of the drugs currently used target GPCRs. GPCRs have a structure with seven cross-membrane domains (for reviews, see [19,39,40,41,42,43]). Once activated by its ligand, a GPCR stimulates the heterotrimeric G protein to which it is coupled. The G protein is composed of three subunits, α, β, and γ, and has GTPasic activity. In its inactive state, the subunit α binds a GDP molecule (guanosine diphosphate) onto its GTPase domain. Its coupling with the receptor leads to the replacement of GDP (guanosine diphosphate) with GTP (guanosine triphosphate), which is responsible for the dissociation of the Gα subunit from the Gβγ dimer. Each of the Gα and Gβγ subunits will trigger the activation of different downstream effectors. The subunit α, via the hydrolysis of its GTP, activates its effectors and returns to an inactive state complexed to the dimer Gβγ. The return to an inactive state of the Gα subunit linked to Gβγ is regulated by the regulator of G protein signaling proteins (RGSs) and G protein-coupled receptor kinases (GRKs) (for reviews, see [19,39,40,41,42,43]). The classification of GPCRs into five classes is based on differences in their structure, ligand binding, and roles in physiology (for reviews, see [19,39,40,41,42,43]). The three major classes are Class A for rhodopsin-like, class B for secretin-like, and class C for melanotrope-glutamate/pheromone receptors (for reviews, see [19,39,40,41,42,43]). Class B GPCRs share low homology with other receptors, such as class A receptors, with which they share only 12% of their sequence homology [41]. Class B GPCRs include the receptors of glucagon like peptide-1 (GLP-1), glucagon like peptide-2 (GLP-2), glucagon, gastric inhibitory peptide (GIP), vasoactive intestinal polypeptide (VIP), and pituitary adenylate cyclase-activating polypeptide (PACAP) [41]. In beta-cells, GPCR activation leads to the stimulation of various signaling pathways such as the cAMP/protein kinase A (PKA)/Exchange proteins activated by cAMP (*Epac*) and the inositol triphosphate (IP_3_)/diacylglycerol (DAG) pathways, as well as to changes in protein phosphorylation and acylation (Figure 2) (for reviews, see [18,19]). The cAMP/PKA/Epac and the IP_3_/DAG pathways engaged following the activation of GPCRs are among the most important signaling pathways for beta-cell’ function, differentiation, and survival [18,19]. Gαs mediates increases in cAMP production through the activation of the adenylate cyclase associated with the potentiation of glucose-induced insulin secretion, while Gαi mediates decreases in cAMP production and the inhibition of insulin secretion [18,19] (Figure 2). Gαq mediates increases in IP_3_ and DAG production through the activation of the phospholipase C associated with the increased release of calcium (Ca^2+^) from the endoplasmic reticulum (ER) and enhanced insulin secretion [18,19] (Figure 2). Moreover, through the activation of multiple signaling pathways such as the extracellular regulated protein kinases 1 and 2 (ERK1/2, or p42/p44 mitogen activated protein kinases (p42/p44 MAPKinases)), phosphoinositide 3-kinase (PI3K), and key transcription factors such as the transcription factor duodenal homeobox 1 (PDX1) and the cAMP response element-binding protein (CREB), GPCRs tightly control beta-cell function, differentiation state, survival, and proliferation [18,19,21,22,23,24]. Current strategies aim to improve the existing therapies targeting beta-cell GPCRs (i.e., GLP-1 and GIP receptors) and develop different beta-cell GPCR ligand combinations and new biased agonists selectively activating the GPCR-initiated signaling pathway (for reviews, see [19,21,22,23,24,44,45,46,47,48,49]) (Figure 2).

#### 2.1.1. Glucagon-Like Peptide-1 Receptor

Oral glucose increases insulin secretion by more than 50% compared to intravenous glucose [21,22,23,24]. This “incretin effect” is due to the secretion of the hormones GLP-1 and GIP by the intestinal endocrine cells during food ingestion [21,22,23,24]. The GLP-1 receptor (GLP-1R) is expressed in a large number of tissues, including the pancreatic beta-, delta-, and possibly alpha-cells; heart; stomach; intestine; brain; and lungs [21,22,23,24,41]. GLP-1 stimulates glucose-dependent insulin secretion from beta-cells and decreases glucagon secretion from alpha-cells [21,22,23,24]. GLP-1 acts in the stomach by decreasing gastric emptying, while in the brain it decreases appetite [21,22,23,24]. GLP-1 stimulates cardiac functions and exerts cardioprotective effects [21,22,23,24]. In muscle and adipose tissue, GLP-1 increases insulin sensitivity and glucose uptake and storage, while it decreases glucose production in the liver [21,22,23,24]. The binding of GLP-1 on its receptor can favor its coupling with various G proteins, Gαs, Gαq, and Gαo, activating several downstream effectors such as adenylate cyclase, phospholipase C, and distinct signaling pathways such as PKA/Epac2, protein kinase C, ERK1/2, and PI3K [19,21,22,23,24,41]. In beta-cells, the GLP-1R is coupled to Gαs, leading to cAMP production, the subsequent activation of the PKA and Epac2 signaling pathways, and potentiation of glucose-induced insulin secretion (Figure 2) [19,21,22,23,24,41]. The activation of the GLP-1R also leads to the activation of downstream signaling pathways such as ERK1/2 and PI3K [19,21,22,23,24,41]. The GLP-1R was further found to induce signaling pathways independent of G proteins but dependent on β-arrestin scaffold proteins (for reviews, see [19,50,51,52,53]). All the pathways activated by GLP-1 binding to its receptor regulate insulin secretion, beta-cell differentiation, and survival. The GLP-1R is a major, well known, and well described therapeutic target for the treatment of T2D (for reviews, see [19,21,22,23,24,41,50,51,52,53]).

#### 2.1.2. Gastric Inhibitory Peptide Receptor

The incretin effect accounts for around 70% of the increase in the insulin response after a meal. GIP is well described to be responsible for most of the incretin effect in healthy individuals [54]. The amount of insulin in circulation following an oral glucose load is ~44% due to GIP, ~22% due to GLP-1, and ~33% due to glucose [55]. Supraphysiological levels of GLP-1 induce an important potentiation of the insulin secretion induced by glucose [56]. GLP-1 is able to stimulate insulin secretion in patients with T2D, while GIP is almost inactive [57]. This observation has led to less interest in targeting the GIP receptor (GIPR) as a therapeutic strategy for T2D [56,57]. However, recent insights into incretin’s physiology and biology revealed that bi-agonist compounds (i.e., GIP and GLP-1) can exert synergistic and robust effects on insulin secretion in human islets [44]. Thus, while GIP alone has no effect on improving glycemic control, GIP in combination with GLP-1 exerts a powerful effect on glycemic and weight control [58]. In beta-cells, the GIPR is coupled to adenylate cyclase through Gαs. The activation of GIPR leads to cAMP production and the potentiation of glucose-induced insulin secretion [19,41,59] (Figure 2). Important studies are currently underway to clarify the role of GIPR in the therapeutic effects of bi-agonist compounds, as well as the signaling networks associated with GIPR activation and G-dependent and G-independent mechanisms involving scaffold proteins.

#### 2.1.3. Glucagon Receptor

Glucagon is produced and secreted by alpha-cells [9,10,11]. Glucagon regulates beta-cell function via paracrine communication and its effects in the islets of Langerhans [9,10,11,60,61,62]. Specific glucagon receptors (GCGRs) are expressed and functional in beta-cells (for reviews, see [60,61,62]). The binding of glucagon to its receptor leads to the activation of the Gαs protein, the production of cAMP, and the potentiation of insulin secretion [41,60,61,62] (Figure 2). Rodriguez Diaz and colleagues confirmed that the glucagon secreted by alpha-cells plays a major role in the regulation of neighboring beta-cells via paracrine communication [63]. However, the recognized importance of hyperglucagonemia in the pathophysiology of T1D and T2D has encouraged the development of therapeutic strategies aimed at reducing the action of glucagon, with some advantages and drawbacks (for reviews, see [61,62,64]). Hence, numerous studies have proposed blocking the effects of glucagon at its receptor to reduce hyperglycemia as a therapeutic strategy to treat diabetes (for reviews, see [61,62,64]). The use of a GCGR antagonist molecule in the db/db diabetic mouse model and in diabetic mice induced by a high-fat diet and streptozotocin improved hyperglycemia and promoted beta-cell regeneration [65]. Blocking the action of glucagon with a monoclonal antibody of GCGR (Ab-4) improved hyperglycemia [65]. Treatment with an antibody of GCGR promoted beta-cell survival, improved the formation of functional beta-cell mass, and produced insulin-positive cells from cell precursors in NOD mice, a mouse model of T1D [66]. The anti-diabetic effect and regenerative action of cells using the GCGR antibody have been confirmed [67]. Combined with the anti-CD3 teplizumab immune modulator, the GCGR antibody significantly increased beta-cell mass and reduced diabetes’ progression in NOD mice [67]. These studies proposed the use of anti-GCGR antibodies as an effective treatment to reduce the progression of T1D.

#### 2.1.4. Melatonin Receptors

The neurohormone melatonin is secreted by the pineal gland in a circadian rhythm, with higher levels observed at night. Melatonin targets two high-affinity GPCRs: the melatonin receptor 1A (also known as MT1, encoded by *MTNR1A*) and MT2 (encoded by *MTNR1B*) [68,69,70,71]. MT1 and MT2 are coupled to Gαi and negatively regulate adenylate cyclase. Genome-wide association studies have revealed that common non-coding variants of *MTNR1B* (encoding MT2) increase the risk of T2D. The *MTNR1B* risk allele genotype associated with increased MT2 expression in pancreatic islets has been suggested to mediate the inhibitory effect of melatonin on beta-cell cAMP levels and insulin secretion [68,69,70,71]. However, various human and animal studies have revealed that the role of melatonin in the regulation of glucose homeostasis is not fully elucidated [68,69,70,71]. Future studies are needed to determine the MT2 receptor expression profile in human tissues and its signaling pathways and physiological effects in order to clarify the link between melatonin and T2D. It is important to address this issue in a context in which melatonin is widely used and available in an increasing number of countries.

#### 2.1.5. Free Fatty Acid Receptors

Activated by long-chain free fatty acids such as palmitate and oleic acid, free fatty acid receptor 1 (FFAR1 or GPR40) stimulates phospholipase C through Gαq coupling [72,73,74] (Figure 2). The activation of FFAR1/GPR40 was shown to potentiate glucose-induced insulin secretion and to stimulate GLP-1 release in small intestinal enteroendocrine cells [75]. Synthetic small-molecule GPR40 full agonists and super-agonists which enhance insulin secretion in a glucose-dependent manner in vitro and in vivo have been described (for reviews, see [72,73,74,75,76,77]). A GPR40 agonist that can be controlled by light (FAAzo-10) has proved a useful tool for studying the effects of fatty acid derivatives on beta-cell function [78]. FFAR1 therefore logically emerged as an interesting therapeutic target for T2D [72,73,74,75,76,77]. The role of free fatty acid receptor 2 (FFAR2 or GPR43) on islet function has also been investigated (for reviews, see [79,80,81]). It is accepted that FFAR2 is coupled to Gαi, inhibiting cAMP production and insulin secretion. It is important to note the existence of contradictory results between species, since some studies using the human beta-cell line or pseudo islets demonstrated that FFAR2 activation inhibited glucose-induced insulin secretion, while other studies indicated its stimulatory effects on insulin secretion using mouse islets [79,80,81]. Hence, studies are still needed to clearly identify the role of FFAR2 in beta-cells and the potential use of FFAR2 modulators. Based on the available data, the use of antagonists of FFAR2 is proposed to be the most logical pharmacological approach to treat T2D [79,80,81]. Another free fatty acid receptor, free fatty acid receptor 4 (FFAR4 or GPR120) was reported to be expressed and functional in beta-cells (Figure 2) [82,83]. The activation of FFAR4/GPR120 stimulates phospholipase C (Figure 2) and Ca^2+^ release from the ER, ERK1/2, and PKB/Akt signaling pathways, regulating insulin secretion and beta-cell survival [82,83]. These findings propose that GPR120 is a potential therapeutic target for T2D [82,83].

#### 2.1.6. Muscarinic Acetylcholine Receptor

The activation of the parasympathetic nervous system stimulates beta-cell function [84]. The main parasympathetic neurotransmitters are acetylcholine, VIP, and PACAP [84]. Muscarinic acetylcholine receptors (M_3_Rs) are well described to be expressed and functional in beta-cells [84,85]. Following their activation, M_3_Rs increase insulin secretion through coupling with the Gαq protein, activating the phospholipase C pathway [84,85] (Figure 2). A positive allosteric modulator (PAM) of M_3_R was reported to promote insulin secretion in mice. This study suggests that PAMs of M_3_R could become novel anti-diabetic agents [85].

#### 2.1.7. Adrenergic Receptors

The activation of the sympathetic nervous system regulates insulin secretion [86,87]. The main sympathetic neurotransmitters are norepinephrine and epinephrine [86,87]. Adrenergic receptors are expressed in beta-cells [86,87]. The β2 subtype adrenergic receptor (β2AR) is primarily coupled to Gαs, leading to cAMP production and the potentiation of glucose-induced insulin secretion [87,88] (Figure 2). β2AR-selective agonists were described to stimulate insulin secretion in human islets [87,88,89]. The α2 subtype adrenergic receptor (α2AR) is also expressed in beta-cells and is found to be coupled to Gαi, inhibiting insulin secretion [90,91] (Figure 2). The antagonists of α2AR were reported to enhance glucose-induced insulin secretion in diabetic patients, revealing a key role of the α2AR in the regulation of the inhibition of insulin release [90,91].

#### 2.1.8. Somatostatin Receptor

Somatostatin, produced and secreted by delta-cells, is well described to regulate beta-cell function via paracrine communication and its effects within the islets of Langerhans [25,26]. Somatostatin acts on the somatostatin receptor (sstr2) coupled to Gαi/ Gαo proteins, inhibiting adenylate cyclase activity and reducing cAMP production [25,26] (Figure 2). Somatostatin inhibits insulin secretion [25,26]. Delta-cells, and the paracrine communication between delta-cells and beta-cells, have been shown to be essential for neonatal survival and normal islet function [92]. Importantly, indirect regulation via delta-cell secretion products (i.e., somatostatin) and the activation of sstr2 on alpha-cells is currently thought to be the main mechanism by which GLP-1 inhibits the secretion of glucagon within the pancreatic islets (for a review, see [93]).

#### 2.1.9. Cannabinoid Receptor

GPR55 is a cannabinoid receptor known to be coupled to Gαq and to be expressed in beta-cells [94,95] (Figure 2). The activation of GPR55 was reported to increase glucose-induced insulin secretion and the expression of anti-apoptotic genes such as Bcl-2 and Bcl-xL, to induce the phosphorylation/activation of the transcription factor CREB crucial for beta-cell survival, and to decrease ER stress-mediated apoptosis [95].

#### 2.1.10. Angiotensin Receptor

The inhibition of the renin-angiotensin system delays the onset of T2D in high-risk individuals [96]. The angiotensin 1 receptor (AT1R) is a GPCR [97]. Upon activation, AT1R generates G protein-dependent and G protein-independent signals, and these transduction systems contribute separately to the effects of AT1R on health and disease [97]. Losartan, a selective AT1R blocker, protects human islets from chronic exposure to high glucose concentrations. The treatment of beta-cells with losartan also improved insulin secretion [98]. Telmisartan, another AT1R blocker, is a drug commonly used in the treatment of hypertension. Telmisartan is thought to exert protective effects against glucolipotoxicity-induced beta-cell apoptosis and to improve insulin secretion by inhibiting oxidative and ER stress [99]. Interestingly, the treatment of db/db mice with a combination of telmisartan and dipeptidyl peptidase-4 (DPP-4) inhibitor (i.e., linagliptin) preserved islet cells’ function and morphology by reducing oxidative stress [100].

#### 2.1.11. Orphan and Other G Protein-Coupled Receptors

The activation of G protein-coupled receptor 119 (GPR119), a class A GPCR expressed in beta-cells and intestinal L cells, stimulates both insulin secretion and GLP-1 release [101,102]. The activation of GPR119 in beta-cells by lysophosphatidylcholine and oleoylethanolamide increases cAMP production and stimulates insulin secretion [103] (Figure 2). Research on GPR119 agonists has been carried out, and structurally diverse small-molecule modulators of GPR119 have been reported [104,105]. Notably, the adhesion protein-coupled receptor 56 (GPR56) has been identified as the most abundant GPCR expressed in islets, suggesting its potential role in islet function [106]. Olaniru and colleagues reported that this GPCR can be activated by its endogenous ligand (i.e., extracellular matrix collagen III) to stimulate insulin secretion [107]. GPR142, a class A member of the GPCR family, was reported to be highly expressed in beta-cells and to play a role in maintaining beta-cell function and potentiating glucose-stimulated insulin secretion [108]. Trace amine-associated receptor 1 (TAAR1) is a GPCR expressed in beta-cells. Following activation, TAAR1 was shown to be coupled to Gαs, potentiating PKA/Epac-dependent insulin secretion (Figure 2), activating ERK1/2, CREB, and inducing beta-cell proliferation [109]. Notably, the olfactory receptor chemosensing mechanism was found to be functional in beta-cells and to regulate insulin secretion [110]. Olfactory receptor isoforms are expressed in islets and in MIN6 beta-cell lines, including OLFR15 and OLFR821 [110,111]. A medium-chain fatty acid in food, octanoic acid, which binds OLFR15, was found to potentiate glucose-stimulated insulin secretion and to improve glucose tolerance in vivo [110,111].

### 2.2. Receptors with Intrinsic Enzymatic Activity Regulating Insulin Secretion and Beta-Cell Survival

#### 2.2.1. Receptor Tyrosine Kinases Regulating Beta-Cell Insulin Secretion and Survival

In humans, receptor tyrosine kinases (RTKs) consist of 58 integral cell surface membrane receptors gathered into 20 classes [112]. In the absence of ligands, RTKs are in a monomeric form composed of three domains: an extracellular ligand-binding domain, a single-pass alpha-helical transmembrane domain, and a cytosolic tyrosine kinase domain (TKD) (Figure 3) [112,113,114]. The binding of a ligand induces dimerization or, sometimes, the oligomerization of RTK monomers, which brings the kinase domains of two RTKs in close proximity. As a result of the dimer or oligomer’s formation, the TKD phosphorylates the tyrosine residues within the TKD of each RTK monomers [112,113,114]. The activation of TKD promotes the recruitment and activation of downstream signaling proteins [112,113,114,115]. The recruitment and activation of these proteins stimulate distinct signaling pathways such as ERK1/2, PI3K, protein kinase B (PKB/Akt), PKA, or PKC [112,113,114,115]. RTKs play pleiotropic functions by regulating cell growth, differentiation, and function [112,113,114,115]. Their role in promoting tumor cell growth is well described and has made this receptor family a major therapeutic target [112,113,114,115]. In beta-cells, several RTKs are expressed and play key roles in the control of insulin production, insulin secretion, beta-cell proliferation, and differentiation. As an example, insulin receptor (IR) and c-Kit regulate the trafficking and fusion of insulin-containing granules (for review, see [20]).

##### Insulin Receptor

Although the IR contains the three canonical domains of RTKs, it differs from other RTKs by the separation of its receptor precursor into α- and β-subunits, which are stably linked by a disulfide bond [20,114,116,117] (Figure 3). Its extracellular domain consists of the whole α-subunit and part of the β-subunit; they represent the ligand-binding site. The β-subunit spans the extracellular insert domain β, transmembrane helix, intracellular juxtamembrane domain, TKD, and C-terminal tail domains [20,114,116,117]. Because the IR has disulfide bonds between its two α-subunits, it is postulated that the IR exists as a covalently linked dimer on the plasma membrane [20,114,116,117]. Once insulin binds to its ligand binding site, then the autophosphorylation of intracellular tyrosine residues ensues [20,114,116,117] (Figure 3). Tyrosine 1150, tyrosine 1148, and tyrosine 1151 are sequentially phosphorylated and are followed by the phosphorylation of other tyrosine residues including tyrosine 953, tyrosine 960, tyrosine 1316, and tyrosine 1322. The phosphorylation of tyrosine 1316 and tyrosine 1322 is thought to promote interactions of the IR with adaptor proteins such as insulin receptor substrates (IRSs) [20,114,116,117]. After IR-IRS interactions, IRS adaptor proteins interact with the p85 regulatory subunit of PI3K, which in turn promotes the production of phosphatidylinositol-3,4,5-trisphosphate (PiP_3_) [20,114,116,117]. PiP_3_ recruits the phosphoinositide-dependent protein kinase-1 (PDK1) and activates PKB/Akt, which, in turn, play a pivotal role by phosphorylating various proteins, leading to glucose uptake, glycogen synthesis, protein and fat synthesis, and gene expression [20,114,117,118] (Figure 3). The ERK1/2 pathway is also induced and crucial for regulating the mitogenic effects of insulin on cell proliferation, growth, and differentiation [118,119] (Figure 3). In beta-cells, the PI3K/Akt pathway activated by the IR was shown to regulate insulin secretion [20]. The in vitro exposure of beta-cells to insulin increases intracellular Ca^2+^ and insulin secretion [120]. A loss of class IA PI3Ks reduces the intracellular Ca^2+^, resulting in impaired insulin secretion [121]. Akt is involved in the phosphorylation of the Rab GTPase-activating protein AS160. The latter is thought to contribute to glucose-stimulated insulin release via IR/IRS-2 signaling [122]. The mechanisms through which IRS-1 and -2 regulate insulin secretion involves the ER [123,124]. In a mouse model of beta-cells, it has been shown that IRS-1 colocalizes with the ER Ca^2+^ ATPase, preventing Ca^2+^ reuptake into the ER and thereby keeping Ca^2+^ in the cytosol [124]. Mice with a homozygous disruption of the Irs1 gene display impaired insulin secretion [124]. Besides beta-cell function, the IR’s signaling controls the beta-cell cycle [125]. Insulin exposure increases the proliferation of mouse beta-cells, and blocking insulin secretion with somatostatin blunts the proliferation induced by hyperglycemia [125]. In addition, beta-cell-specific knock out mice of IR or IRS-2 hampered beta-cell replication [126], which was associated with the reduced expression of centromere protein A (CENP-A) and polo-like kinase 1 (PLK1). PLK1 is a kinase that regulates mitotic entry and exit [127]. CENP-A is a centromere-specific histone H3 variant that is involved in mitotic progression and chromosome segregation in mammalian cells [128]. A beta-cell-specific CENP-A knockout in mice resulted in reduced adaptive beta-cell proliferation to the insulin resistance induced by aging, pregnancy, a high-fat diet, and insulin receptor antagonist S961 [129].

##### Insulin Like Growth Factor-1 Receptor

The insulin like growth factor-1 receptor (IGF-1R) displays a membrane-spanning tetrameric structure. It is synthesized as a single-chain α-β pro-receptor and processed by proteolysis and glycosylation [130]. In its mature and functional form, IGF-1R consists of two identical extracellular α-subunits and two identical β-subunits, all linked by disulfide bridges. The β-chain contains an extracellular domain, a transmembrane domain, and a kinase domain [130,131] (Figure 3). There is high homology (70%) between the IGF-1R and the IR amino acid sequences [131]. The kinase domain is highly homologous to that of the IR (84%), its juxtamembrane domain shares 61% of its homology with the IR, whereas its C-terminal domain shares only 44% [131] (Figure 3). Despite this high degree of homology, these two receptors have distinct biological roles. IGF-1R is a potent regulator of cell growth, proliferation, and differentiation [131]. Following binding to its ligand, the insulin like growth factor-1 (IGF-1), IGF-1R activates signaling pathways, including the PI3K/PKB/Akt pathway, which enhances beta-cell survival and proliferation. This is critical for maintaining adequate beta-cell function [132,133]. IGF-1R signaling positively influences insulin secretion [133]. The activation of IGF-1R can enhance glucose-stimulated insulin secretion, partly by improving beta-cell function and responsiveness to glucose [132,133]. IGF-1R signaling is proposed to be implicated in beta-cell regeneration and proliferation. IGF-1R activation may help in the regeneration of beta-cells, thereby supporting its role in the restoration of a functional beta-cell mass (for reviews, see [132,133,134]).

##### Epithelial Growth Factor Receptor

Epithelial growth factor receptor (EGFR) is a member of the ErbB receptor family which consists of four transmembrane RTKs [115,135]. The phosphorylation of EGFR’s tyrosine residues is induced by ligand binding, which, in turn, initiates a number of downstream signaling cascades [135]. EGFR is expressed in pancreatic islets [136]. This receptor has been reported to be essential in beta-cell development, function, and proliferation [136,137,138,139,140,141,142,143]. A global EGFR knock-out resulted in delayed beta-cell differentiation and impaired beta-cell proliferation during organogenesis [136]. Moreover, the inhibition of EGFR signaling in the pancreas led to impaired beta-cell proliferation postnatally, during pregnancy, and in response to a high-fat diet [138]. The beta-cell-specific depletion of EGFR results in an impaired expression of cyclin D1 and an impaired suppression of p27 after a partial pancreatectomy, both of which enhance beta-cell proliferation [143]. These data highlight the importance of EGFR signaling and its downstream signaling pathways in postnatal beta-cell growth.

##### C-Met Receptor

C-Met receptor is a TKR that is composed of a semaphorin (SEMA) domain, a plexin-semaphorin-integrin (PSI) domain, four consecutive immunoglobulin-plexin-transcription factor (IPT1–4) domains in its extracellular region, a single transmembrane helix (TM), and an intracellular kinase domain (KD) [144]. C-Met is initially expressed as a 150 kDa single-chain precursor which then turns into its mature form through the proteolytic cleavage between Arginine 307 and Serine 308 by the furin protease [144]. Its mature form contains an α- and β-subunit that are linked by at least three disulfide bonds. C-Met is activated by its only cognate ligand—hepatocyte growth factor (HGF) [145,146]. The binding of HGF to c-Met induces the dimerization of c-Met that enables its intracellular KD to undergo autophosphorylation in tyrosine residues (i.e., tyrosine residues 1234 and 1235) [147]. These tyrosine residues provide docking sites for adaptor proteins including the growth factor receptor-bound protein 2 (Grb2), son of sevenless (SoS), Grb2-associated adaptor protein (GAB1), SRC homology protein tyrosine phosphatase 3 (Shp2), PI3K, and the signal transducer and activator of transcription 3 (STAT3) [148,149]. In addition, GAB1, once bound to and phosphorylated by c-Met, further creates binding sites for more downstream adaptors [148,149]. This leads to the activation of various downstream signaling pathways such as ERK1/2, Jun amino-terminal kinases (JNK1, JNK2 and JNK3), p38 MAP kinase, PI3K/PKB/Akt, and the signal transducer and activator of transcription (STAT) pathways. The activation of c-Met controls cell proliferation, survival, migration, and invasion [150]. As an example, the activation of c-Met induces PI3K/PKB/Akt stimulation, mediating cell survival and resistance to apoptosis through the inactivation of the pro-apoptotic protein BCL-2 antagonist of cell death (BAD) and the degradation of the pro-apoptotic protein p53 [151]. Notably, the receptor c-Met is expressed and functional in beta-cells, and, with HGF, it plays a key role in the control of beta-cell mass, regeneration, and adaptive response to insulin resistance. In mice, c-MET promotes beta-cell survival and proliferation in response to insulin resistance [152,153,154,155,156]. This role could be key to beta-cell mass and function adaptation during obesity and pregnancy [153,154]. In mice, the beta-cell-specific suppression of the c-Met receptor leads to a reduction in plasma insulin levels caused by an increase in beta-cell death and a subsequent reduction in beta-cell mass. Glucose intolerance and impaired insulin secretion are aggravated in pancreatectomized (PX) mice who have a beta-cell-specific ablation of their c-Met receptor compared to wild-type PX mice [155].

#### 2.2.2. Receptor Serine/Theonine Kinase Regulating Beta-Cell Insulin Secretion and Survival

The transforming growth factor-β (TGF-β) receptor family consists of type 1 receptor (TGFBR1), also known as activin receptor-like kinases (ALKs), the transmembrane type 2 receptor (TGFBR2), and type 3 receptors (TGFBR3) [157,158]. TGFRB1 and TGFRB2 receptors form a hetero-tetrametric complex that phosphorylates a specific set of intracellular SMAD proteins in response to ligands including TGF-β1, 2, and 3 [158]. TGFBR2 binds the TGF-β ligands, resulting in the recruitment and phosphorylation of TGFBR1. In vascular endothelia where betaglycan and endoglin are the most abundant, the two TGFBR3s can either facilitate the binding of TGF-β to TGFBR2 or sequester the TGF-β ligand to suppress TGFBR1 receptor signaling [159]. In cases where TGF-β binds to and activates TGFBR2, the latter phosphorylates TGFBR1, which, in turn, phosphorylates the cytoplasmic receptor-regulated Smads (R-Smads) including Smad2 and Smad3. Then, phosphorylated Smad2/3 interact with Smad4, also called the common Smad or co-Smad, forming a trimeric complex which translocates into the nucleus for regulating the transcription of target genes [160]. In addition, TGF-β signaling also contains an inhibitory Smad protein, Smad7, to counterbalance the signal strength via a negative feedback mechanism [161]. The signaling induced by the TGFBR heterodimeric complex plays a key and distinct role in basal and adaptive beta-cell proliferation. In mice, partial duct ligation (PDL) promotes local islet inflammation, which, in turn, stimulates beta-cell proliferation as an adaptive mechanism, leading to increased beta-cell mass [162]. In the PDL model, TGF-β1 ligands released by M2 macrophages could account for the beta-cell proliferation induced by PDL [163]. The beta-cell proliferation induced by M2 macrophages relies on the stimulation of Smad7 [163]. In the PDL mice model, the islet-specific deletion of both Tgfbr1 and 2 abrogates the beta-cell proliferation induced by inflammation [162]. Unlike this model, the TGFBR heterodimeric complex plays a negative role in the beta-cell proliferation induced by a partial pancreatectomy (PPX). In PPX mice with an islet-specific deletion of both Tgfbr1 and 2, beta-cell proliferation was enhanced when compared to PPX wild-type mice [162]. TGFBR could play different roles according to different external signals. The TGFBR heterodimeric complex is also involved in beta-cell dedifferentiation, which is characterized by the loss or decreased expression of beta-cell marker genes including insulin, the glucose transporter (GLUT2), the MAF BZIP transcription factor (Mafa), and the PDX1 transcription factor. As a result, beta-cells are unable to secrete insulin in response to glucose [164,165,166]. The dedifferentiation of beta-cells occurs in human islets that are in vitro cultured for a long time, thereby mimicking T2D [165,166]. In this model, dedifferentiation relied on the activation of TGFBR1 [164]. The suppression of TGFBR1 by shRNAs abolishes the dedifferentiation and proliferation of cultured human beta-cells [164]. In T2D, beta-cell dedifferentiation is associated with oxidative stress. TGF-β1 increases reactive oxygen species by inhibiting the activity of complex IV of the respiratory chain and reducing the expression of intracellular antioxidants including glutathione, superoxide dismutase, catalase, and glutaredoxin [167,168]. While the stimulation of TGFBR could contribute to beta-cell dedifferentiation and oxidative stress, the activation of that signaling can be protective against islet inflammation. Indeed, the overexpression of TGF-β1 in the mouse model of T1D prevents autoimmune beta-cell injury [169,170]. Mice with a Tgf-β1 1 knockout or T cell-specific deletion of Tgfbr2 develop multi-organ autoimmune injury [171,172]. Likewise, abnormal immunity, although less severe, was also observed in Smad3-null mice, which is also associated with dysregulated T cell activation [173].

#### 2.2.3. Cytokine Receptor Regulating Beta-Cell Insulin Secretion and Survival

Pro-inflammatory cytokines have been well described to induce apoptosis and dysfunction through the activation of membrane receptors and subsequent intracellular protein kinases, leading to mitochondrial alterations and the release of death signals, ER stress, and the regulation of gene transcription [2,174] (Figure 4). Mitogen-Activated Protein Kinases such as JNKs are activated by pro-inflammatory cytokines and trigger the release of mitochondrial death and ER stress signals [2,174]. The nuclear factor-kappa B (NFκB) transcription factor is well known to mediate the deleterious effects of pro-inflammatory cytokines [175,176]. The active transcriptional subunit p65 of NFκB is sequestered and inactivated in the cytosol by the inhibitor of NFκB (IκB). Exposure to IL-1β and TNF-α induces a phosphorylation-dependent degradation of IκB through the activation of the kinase IκB (IKKβ), promoting the nuclear translocation of the active transcriptional subunit p65 and the subsequent transcription of pro-inflammatory genes [175,176]. The inhibition of these signaling pathways through the use of inhibitors represents a key strategy to prevent the apoptotic death and dysfunction of beta-cells induced by pro-inflammatory cytokines in both types of diabetes, and in the context of islet transplantation (for reviews, see [174,177]).

##### Interleukin-1β Receptor

The type I IL-1β receptor (IL-1R) is responsible for mediating the inflammatory effects of IL-1β [2,174]. The type II IL-1R can act as a suppressor of IL-1β activity by competing for IL-1β binding [2,174]. The IL-1 receptor antagonist (IL-1RA) blocks the effects of IL-1β [2,174]. The IL-1R antagonist anakinra, approved for the treatment of rheumatoid arthritis, has been proposed to prevent autoimmune beta-cell destruction in patients with T1D. Randomized controlled trials of the IL-1R blockade have been carried out with anakinra [178]. Anakinra proved ineffective as a single immunomodulatory drug in new-onset T1D [178]. In contrast, an anakinra administration exerted anti-diabetic effects in combination with anti-CD3 antibodies in diabetic NOD mice, suggesting that combined anakinra/anti-CD3 antibody therapy could be a therapeutic strategy for patients with T1D [179]. The IL-1 receptor antagonist has shown promise in clinical trials for the treatment of T2D [180]. Treatment with anakinra improved blood glucose levels by increasing the insulin secretory function of beta-cells [180]. Beta-cells have been proposed as a potential source of IL-1β, which is produced as inactive pro-IL-1β and converted to biologically active IL-1β by inflammasome-mediated enzymatic cleavage [181]. This secretion of IL-1β induces beta-cell apoptosis (for reviews, see [37,38,180,181]). It has also been proposed and tested to block the deleterious actions of IL-1β using IL-1β-specific antibodies. Using canakinumab, a human anti-IL-1β monoclonal antibody, a randomized controlled trial of an IL-1β blockade was carried out to investigate a possible improvement in the beta-cell function of patients with new-onset T1D [178]. Canakinumab has not been shown to be effective as an immunomodulatory drug in new-onset T1D. The authors proposed that an IL-1β blockade would be more effective in combination with treatments targeting adaptive immunity [178]. The administration of a rat-specific IL-1β monoclonal antibody to the diabetic Cohen rat, a genetic model of nutritionally induced diabetes when fed a sucrose-rich diet, counteracted beta-cell dysfunction and glucose intolerance [182]. A randomized controlled trial of an IL-1R blockade with canakinumab has also been carried out to investigate possible improvements in cardiovascular events and T2D. Treatment with canakinumab had similar effects on major cardiovascular events in diabetic and non-diabetic patients, but did not reduce the number of cases of diabetes [183]. Other anti- IL-1β antibodies such as gevokizumab or LY2189102 have shown promise in clinical trials for the treatment of T2D [184,185]. However, no anti-IL-1β antibody-based treatment for diabetes has yet been approved. Despite the data obtained from preclinical and clinical studies, further research and knowledge are still needed to develop this type of therapeutic strategy for diabetes.

##### Tumor Necrosis Factor-α Receptor

Tumor necrosis factor-α (TNF-α) exerts toxic effects on beta-cells [186]. Patients with T1D typically have elevated serum levels of TNF-α [187]. Golimumab, a human monoclonal antibody specific to TNF-α, was administrated to determine whether the anti-TNF-α antibody could preserve beta-cell function in young people with newly diagnosed overt T1D. In a phase 2, multicenter, double-blind, parallel-group clinical trial, the golimumab treatment promoted an increase in insulin production and a decrease in exogenous insulin use in children and young adults with newly diagnosed T1D [188]. Therapy using an anti-TNF-α antibody, alone or in combination with a T-cell receptor (TCR)-specific antibody, has been suggested to prevent the emergence of T1D. In the T1D LEW.1AR1-iddm rat model, the combination of these two antibodies increased cell proliferation, reduced apoptosis, and led to a restoration of beta-cell mass and function which was associated with a decrease in immune cell infiltration [189].

##### Interferon Receptors

Type I IFN and IFN-α were found to be key cytokines present in the islets of patients with T1D, contributing to the triggering and amplification of autoimmunity [190]. Type I IFN was shown to be expressed in human islets isolated from donors with T1D [191]. Children genetically at-risk for T1D present a type I IFN transcriptomic signature in their circulation prior to the development of auto-antibodies [192]. In human islets, exposure to IFN-α induces the release of chemokines, ER stress, human leukocyte antigen (HLA) class I overexpression, and apoptosis when combined with IL-1β [193]. The action of IFN-α in human beta-cells was found to be mediated by the tyrosine kinase 2 (TYK2) and the signal transducer and activator of transcription 1 and 2 (STAT1 and STAT2) [193]. Inhibitors of TYK2 were developed using structure-based drug design [194]. The inhibitors designed proved to be allosteric inhibitors of TYK2 kinase activity with high specificity for TYK2 [194]. Human beta-cells were protected against the deleterious effects of IFN by treatment with these TYK2 inhibitors [194]. These studies justify future clinical trials on the treatment of T1D using TYK2 inhibitors.

A selective inhibitor of mammalian janus kinase 1 (JAK1) (i.e., ABT 317) reduced the signaling and deleterious effects of IFN-γ in beta-cells [195]. NOD mice treated with ABT 317 were protected against the development of diabetes [195]. This class of inhibitors has therefore been proposed for the treatment of T1D [195]. Immune checkpoint inhibitors (i.e., antibodies that block the interaction between programmed death receptor 1 (PD-1) and programmed death receptor ligand 1 (PD-L1)) are used in the treatment of cancers [196]. Immune checkpoint inhibitor therapies are described as causing immune-related adverse effects, and T1D has been reported as a side effect of immune checkpoint inhibitor treatments for cancer [197]. The administration of a JAK1/JAK2 inhibitor (LN3103801) slowed the development of anti-PD-L1-induced diabetes in NOD mice by blocking IFN-γ’s function on beta-cells and common gamma chain cytokine activities in T cells, providing preclinical validations for the use of JAK inhibitors in checkpoint inhibitor-induced diabetes [198].

##### Other Cytokine Receptors

Importantly, many cytokine receptors are functionally expressed in beta-cells. The activation of some of these receptors is detrimental to beta-cell function and survival (i.e., IL-1β, TNF-α, and interferon receptors). Conversely, certain cytokines such as interleukin-13 (IL-13) and interleukin-6 (IL-6), through the activation of their specific receptors, protect beta-cells against apoptotic death [199,200,201]. For example, IL-13 has been reported to protect human and rat beta-cells against cytokine-induced death. The protective effects of IL-13 were mediated by the IRS2/Akt signaling pathway [199]. By preserving beta-cell survival and mass, IL-13 may be useful for the treatment of T2D [199]. Beta-cells express functional IL-6 receptors [200]. IL-6 protects human beta-cells and islets from death by coupling autophagy to the antioxidant response, reducing reactive oxygen species [200]. Interestingly, within the islet, IL-6 increases GLP-1′s production and secretion by alpha-cells, leading to insulin secretion by beta-cells [201]. Targeting the IL-6 receptor is suggested as a potential therapeutic strategy for T2D.

## 3. Intracellular Receptors

### 3.1. Cytosolic Receptors Regulating Beta-Cell Insulin Secretion and Survival

#### 3.1.1. Steroid Hormone Estrogen Receptors

Notably, three estrogen receptors (ERs), ERα, ERβ, and the G-protein-coupled ER (GPER), were identified and described in human and rodent beta-cells [202,203,204,205,206]. 17β-estradiol (E2) was shown to protect beta-cells against gluco-lipotoxicity, oxidative stress, amyloid polypeptide deposit toxicity, and apoptosis in diabetic rodent models. Key advances in our understanding of the role of ERs in beta-cell function, survival, and proliferation, and in nutrient homeostasis, have been made over the last ten years (for reviews, see [203,204,205]). ERs are proposed as potential key therapeutic targets for the preservation of insulin secretion and beta-cell survival in humans (for reviews, see [203,204,205]). Notably, the efficacy of a GLP-1 and estrogen dual agonist (GLP1-E2) in pancreatic islet protection was recently reported [206].

#### 3.1.2. Vitamin D Receptor

The dysfunction of beta-cells has been shown to be associated with low blood levels of 25-hydroxyvitamin D [207]. Epidemiological and human genetic studies have revealed a link between vitamin D and the vitamin D receptor (VDR), on the one hand, and T1D and T2D, on the other [208]. VDR binds to bromodomain-containing protein 9 (BRD9 protein). The pharmacological inhibition of BRD9 promotes the activation of a VDR-dependent transcriptional program which favors beta-cell survival. The activation of VDR signaling using a synthetic ligand in combination with BRD9 inhibition partially restored insulin secretion in db/db mice and in diabetic mice treated with low-dose streptozotocin [209]. Low blood levels of 25-hydroxyvitamin D have been reported as a risk factor for T2D. Consequently, vitamin D supplementation has been proposed as a potential therapeutic strategy to reduce the risk of diabetes [210,211]. However, in a multicenter, randomized trial of individuals at high risk of T2D and not selected for vitamin D insufficiency, vitamin D3 supplementation did not result in a significant reduction in the risk of diabetes [212]. In the Tromsø Vitamin D and T2D trial, which randomly assigned adults with prediabetes to vitamin D3 or placebo, the risk of diabetes was lower in the vitamin D group than in the placebo group, but the difference was not significant [213]. In clinical trials in which adults with prediabetes were randomly assigned to an active form of a vitamin D analog (eldecalcitol) or placebo, the risk of diabetes was also found to be lower in the vitamin D group than in the placebo group, but the difference was not significant either [214,215].

### 3.2. Ion Channel Receptor Regulating Insulin Secretion and Beta-Cell Survival

The role of the voltage-dependent anion channel-1 (VDAC1), an ATP-conducting mitochondrial outer membrane in T2D, has been investigated in pancreatic islets from organ donors with T2D. The expression of VDAC1 is shown to be increased in islets from patients with T2D. The overexpression of VDAC1 results in its defective targeting of the cell plasma membrane, with a loss of the metabolic coupling factor ATP. Metformin and its specific VDAC1 inhibitor (VBIT-4) were shown to inhibit VDAC1 conduction and restore ATP production and glucose-stimulated insulin secretion in islets. The treatment of db/db mice with VBIT-4 reduced the emergence of hyperglycemia and maintained their beta-cell function and glucose tolerance [216].

### 3.3. Nuclear Receptors Regulating Beta-Cell Insulin Secretion and Survival

#### Glucocorticoid Receptor

Glucocorticoids (GCs) are stress hormones involved in energy regulation, widely used in clinic for their anti-inflammatory, immunosuppressive, and anti-allergic properties [217,218]. Secreted into the systemic circulation, GCs exist in two forms: cortisol (or corticosterone in rodents), which is predominantly bound to the CBG (Corticosteroid-binding globulin), and cortisone (Dehydrocorticosterone in rodents), the mainly free, inactive form [217,218]. At physiological levels, glucocorticoids (i.e., corticosterone and cortisol) were reported to enhance cAMP production in beta-cells, maintaining insulin secretion in the face of perturbed ionic signals [219]. Nevertheless, chronically administered glucocorticoids are described to be diabetogenic [217]. Unlike cortisol, synthetic glucocorticoids used in therapeutics circulate unbound to the CBG. The active form of cortisol (i.e., released from the CBG) passively diffuses across the plasma membrane to bind its receptor (Glucocorticoid receptor (GR)). Once inside the cell, the bioavailability of cortisol is controlled by two enzymes: 11β-HSD2 (11β-hydroxysteroid dehydrogenase type 2), which oxidizes cortisol into cortisone (an inactive metabolite), and 11β-HSD1 (11β-hydroxysteroid dehydrogenase type 1), which has the opposite effect. In its inactive state, the glucocorticoid receptor (GR) is cytoplasmic and associated with proteins such as chaperones (hsp90, hsp70, and p23), and immunophilins of the FK506 family (FKBP51 and FKBP52) [217,218]. Ligand binding to the receptor triggers the dissociation of these protein complexes, releasing two nuclear translocation signals from the GR which translocate into the nucleus to regulate multiple gene expressions [217,218]. GCs exert their diabetogenic effects by inducing or aggravating insulin resistance in target tissues and by exerting deleterious effects on beta-cells [217]. The harmful effects of GCs on insulin secretion were found to rely on the reduction in the expression levels of GLUT2 and glucokinase [220]. Hence, GCs reduce ATP synthesis and limit K^+^/ATP channel closure and membrane depolarization, thus blocking calcium influx into beta-cells [221]. In addition, GCs were shown to increase the expression of proapoptotic factors such as Bax and, conversely, to decrease the expression of anti-apoptotic proteins such as Bcl-2, as well as the phosphorylation of BAD [222]. The inhibition of PKB by GCs and the increased expression and nuclear localization of the proapoptotic transcription factor forkhead box protein O1 (FOXO1) are part of the pathways inducing apoptosis in beta-cells [222]. GCs also activate the JNK and p38 MAP kinase pathways [222]. The activation of these pathways could lead to increased levels of reactive oxygen species, and oxidative stress [222].

## 4. Conclusions

The successful clinical use of numerous drugs targeting membrane or intracellular receptors has revealed the important role of these receptors and of their downstream signaling networks in physiologic and pathogenic cellular processes. Preserving the function and survival of beta-cells in order to achieve long-term glycemic control and prevent complications is an essential feature for an innovative drug to have clinical value in the treatment of diabetes. However, the effectiveness of the current available treatments decreases over time, and their administration presents risks of effects that degrade the quality of life of the patient. In addition, none of the anti-diabetic drugs currently in clinical use promote the maintenance of a functional beta-cell mass over time, revealing an unmet medical need. In this context, innovative research is developing therapeutic strategies to prevent pathogenic mechanisms to protect beta-cells from the deleterious effects of inflammation and/or chronic hyperglycemia over time. In beta-cells, numerous receptors of different structures and classes are expressed which are linked to highly diverse, complex, and interconnected signaling pathways that control their function and/or survival. To date, only one receptor, the GLP-1R, has been targeted for the treatment of T2D. In 2022, another receptor, the GIPR, was targeted for the treatment of T2D with a GLP-1R/GIPR dual agonist. This review presents at least ~30 different receptors currently known to play a role in regulating beta-cell function/dysfunction and/or survival/death. It has now been well described that cellular signaling pathways engaged by receptor activations operate more like webs than superhighways. There are multiple redundancies or alternate routes that can be engaged in response to the activation or inhibition of a pathway. Moreover, signaling pathway crosstalks/interconnections between receptors of different structures and classes, such as GPCRs and TKRs, have been described. Drug efficacy is crucially dependent on the quantity and quality of the receptor(s) targeted, as well as on the state of the signaling pathways and networks in diseased target cells. It is therefore important to map and to understand the behavior of these receptors, signaling pathways, and networks in the diseased beta-cell as comprehensively and accurately as possible. In-depth knowledge of these receptors and their associated signaling networks in healthy beta-cells is also necessary to understand the changes and possible plasticity of these networks in the diseased beta-cell. This comprehensive map of expressed receptors and interconnected signaling networks, as well as their changes and plasticity in healthy and diseased human beta-cells, has not yet been achieved. Highly sophisticated, large-scale, and chemical analyses combined with the use of bioinformatics and artificial intelligence could be designed to establish a comprehensive map of the receptors and interconnected signaling pathways and networks in healthy and diseased human beta-cells. This research should shed light on why current treatments targeting beta-cells may prove ineffective over time and should reveal important new therapeutic targets for preserving the long-term function and survival of beta-cells in patients with diabetes.

## Figures and Tables

**Figure 1 cells-13-01244-f001:**
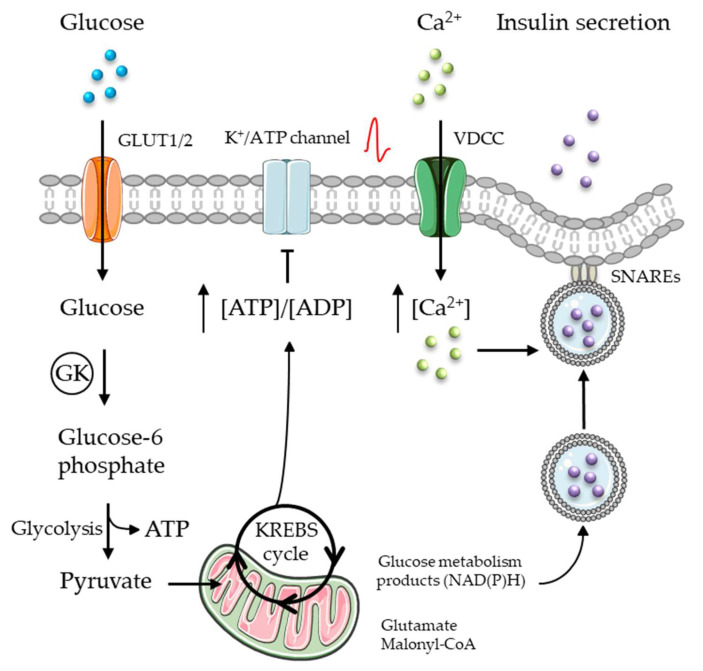
Insulin secretion induced by glucose in the beta-cell. Insulin secretion follows variations in plasma glucose concentrations. Glucose enters the beta-cells through glucose transporters (GLUT1/2). The stimulatory effect of glucose on insulin secretion depends on its metabolism in beta-cells, through glycolysis in the cytosol and KREBS cycle in the mitochondria, and is mediated by triggering and amplifying pathways. The increase in cytosolic ATP concentration ([ATP]) causes the closure of membrane K^+^/ATP channels, leading to membrane depolarization, and the opening of voltage-dependent calcium channels (VDCCs). The entry of calcium triggers the insulin secretion through a soluble N-ethylmaleimide-sensitive factor attachment protein receptor (SNAREs)-mediated fusion of a readily releasable pool of insulin-containing vesicles within the plasma membrane. This triggering pathway is responsible for the first phase of insulin secretion (or the early peak of insulin, 5–10 min). The second phase, also known as the amplifying pathway, is late, more sustained (30–60 min), relies on K^+^/ATP channel-independent mechanisms, and involves several metabolites including glucose metabolism products (NAD(P)H), and associated products (Glutamate and Malonyl-CoA).

**Figure 2 cells-13-01244-f002:**
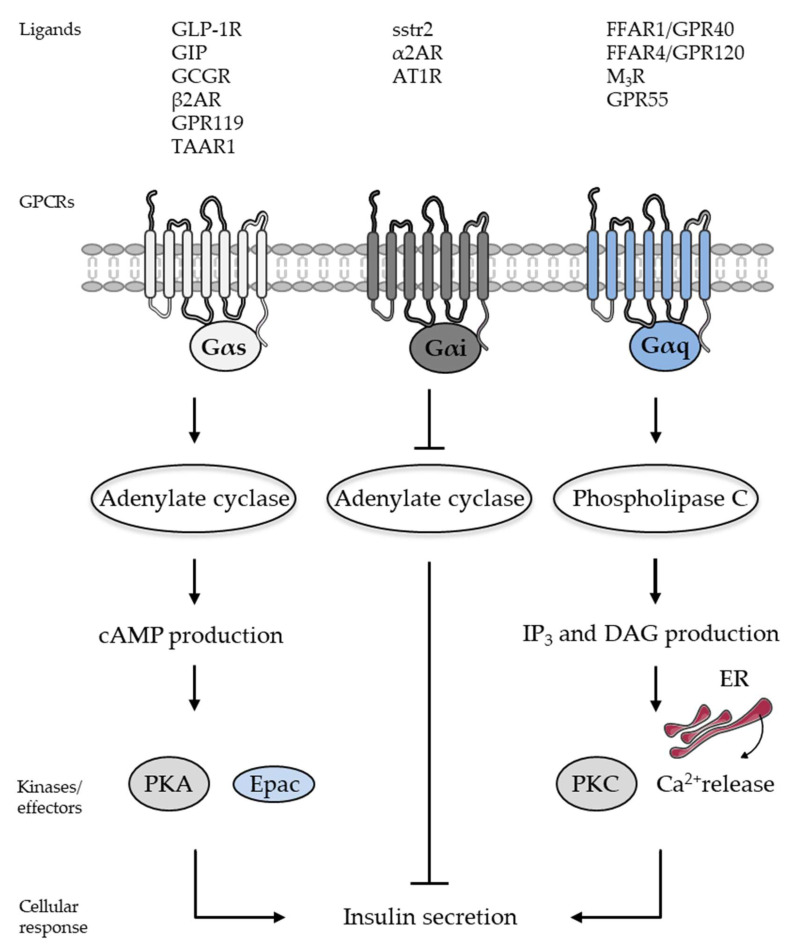
GPCRs regulating beta-cell function. GPCRs stimulate or inhibit insulin secretion in beta-cells. GPCRs coupled to Gαs lead to cAMP production through adenylate cyclase, the subsequent activation of PKA and Epac signaling pathways, and the potentiation of insulin secretion. GPCRs coupled to Gαq lead to inositol triphosphate (IP_3_) and diacylglycerol (DAG) production through phospholipase C, the activation of PKC and calcium (Ca^2+^) release from the endoplasmic reticulum (ER), and the potentiation of insulin secretion. GPCRs coupled to Gαi inhibit insulin secretion by preventing cAMP production.

**Figure 3 cells-13-01244-f003:**
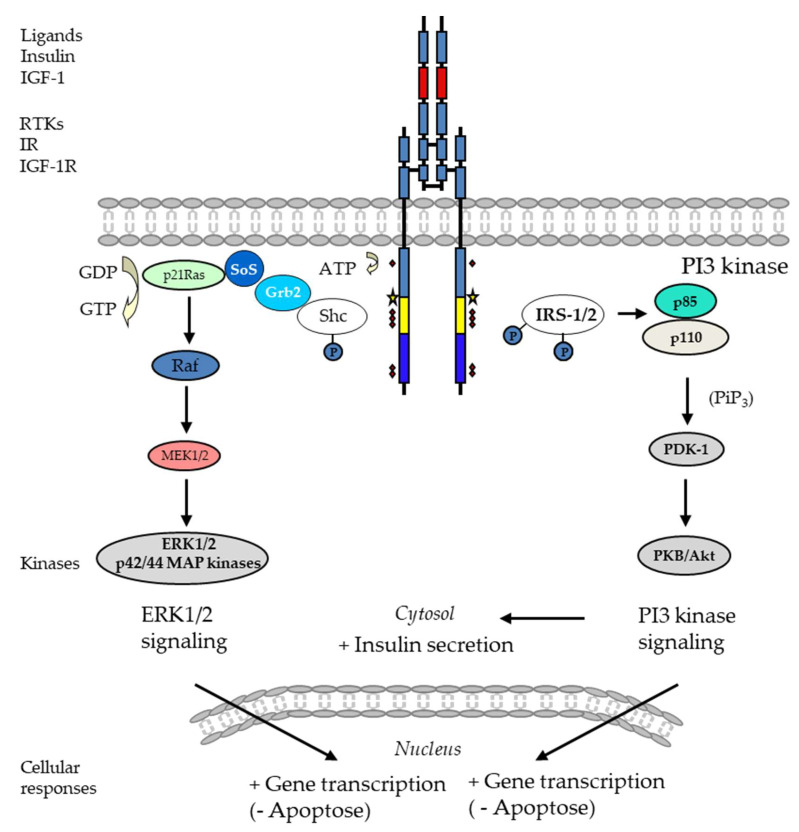
Insulin and IGF-1 receptors regulating beta-cells. Adaptor proteins of the insulin receptor (IR) and insulin like growth factor-1 receptor (IGF-1R) are the insulin receptor substrates 1/2 (IRS-1/2), src homology domain containing proteins (Shc), growth factor receptor-bound protein 2 (Grb2), and son of sevenless (SoS). Following activation by their ligands, IR and IGF-1R interact with these adaptor proteins. Once bound to and phosphorylated by the receptor, the adaptor proteins engage downstream and distinct signaling pathways. Following IR-IRS associations, IRS adaptor proteins interact with the p85 regulatory subunit of PI3 kinase, activating the catalytic subunit p110, which in turn promotes the production of phosphatidylinositol-3,4,5-trisphosphate (PiP_3_). PiP_3_ recruits the phosphoinositide-dependent protein kinase-1 (PDK1), and activates protein kinase B (PKB/Akt). Following IR-Shc/Grb2 association, the ERK1/2 pathway is also induced. Diamonds and star symbols mark the positions of the important tyrosine residues in the transmembrane β subunits, and which are phosphorylated following receptor activation by their ligand.

**Figure 4 cells-13-01244-f004:**
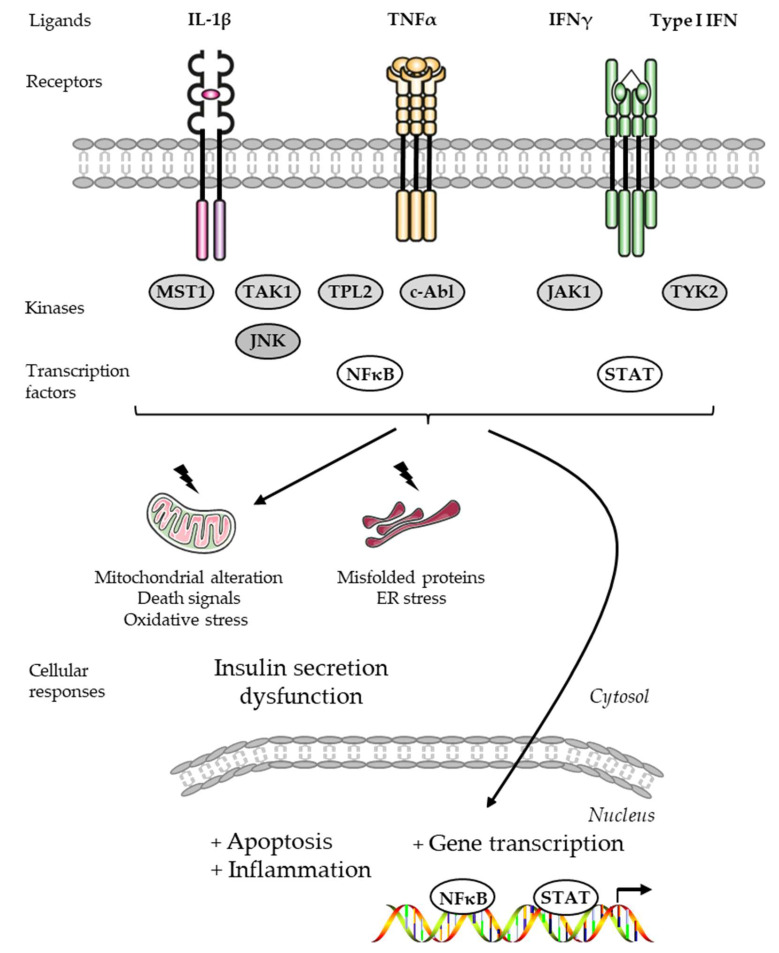
Cytokine receptors regulating beta-cell function and survival. Pro-inflammatory cytokines such as interleukin-1β (IL-1β) and tumor necrosis factor-α (TNF-α) induce the apoptosis and dysfunction of beta-cells through the activation of membrane receptors, intracellular protein kinases (i.e. serine/threonine kinase mammalian sterile 20-like kinase 1 (MST1), serine/threonine kinase transforming growth factor-β activated kinase-1 (TAK1, or MAP3kinase 7), tumor progression locus 2 kinase (TPL2, or MAP3kinase 8), cellular abelson tyrosine kinase (c-Abl), mitogen-activated protein kinases (MAPKs) such as c-Jun N-terminal kinases (JNK)), and nuclear factor-kappa B (NFκB) transcription factor. IFNγ and type I IFN induce the apoptosis and dysfunction of beta-cells through the activation of the membrane receptors, mammalian janus kinase 1 (JAK1) and tyrosine kinase 2 (TYK2), respectively, and signal transducer and activator of transcription (STAT). These signaling pathways lead to mitochondrial alterations and the release of death signals, endoplasmic reticulum (ER) stress, and the regulation of gene transcription.

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
