# Peer review of "Receptors and Signaling Pathways Controlling Beta-Cell Function and Survival as Targets for Anti-Diabetic Therapeutic Strategies"

_cells, 2024, doi:10.3390/cells13151244_

Round 1

Reviewer 1 Report

Comments and Suggestions for Authors

The Review article from Dalle and Abderrahmani focus of the role of receptors and signaling pathways that could be of interest for developing novel anti-diabetic therapeutic approaches. The review is quite extensive and thorough. There are minor aspects that can improve the legibility of the article:

1)        In some part the text can be shortened because it contains redundant portion or not relevant statements. Below are few examples:

a.        Page 2, lines 45-49 (….including…..1,000 endocrine cells) could be removed.

b.       Page 4, line 115. T2D represents nearly 90% of forms of diabetes in humans. This has been already stated in the text.

c.        Page 4, line 126-127. Thus, in T1D….insulin insufficiency. This has been already stated in the text.

d.       Page 4, lines 143-145. Researchers and clinicians….pathways that leads to diabetes. This is redundant with respect the previous sentence.

e.        Page 8, lines 322-323. Importantly, little or no….on alpha-cells. Is this relevant?

f.          Page 10, lines 379-385. With this dogmatic…receptors (VEGFRs). Is this relevant?

2)        Page 1, lines 33-34. The authors state that “diabetes is classified into fours types” but cite only T1D and T2D.

3)        Page 3, line106. T1 should be T1D.

4)        Page 6, line 233. The authors state “Unfortunately, high levels of GIP do not exert effect similar to those of GLP-1”. This statement contradicts what on line 229.

5)        Page 6, line 248. GcgR or GCGR?

6)        Pay attention to some French terms throughout the text!! Adrenergic no adrenergique, Apoptosis not Apoptose.

Overall with these minor correction the review is worth publication.

Comments on the Quality of English Language

Publish after minor revisions

Reviewer 2 Report

Comments and Suggestions for Authors

This review provides a very high-level look at the different receptors and signalling pathways which modulate beta-cell function and viability. There is an appropriate level of detail in most areas, although some omissions are noted which I think should be included (see below). Overall, I think that this review is well put together, fairly comprehensive and will be a helpful addition to the literature in the area.

Some minor comments:

Line 109: Some acknowledgement should be given that T1D is a heterogenous disease with at least two disease endotypes identified (T1DE1 & T1DE2). Whilst those diagnosed early in life follow the more textbook pattern of immune cell mediated beta-cell destruction, those diagnosed later have less immune cells infiltrating the islet, less beta-cell destruction and many more residual insulin containing islets despite low/absent insulin secretion.

Section 2.1 There is no section included for the melatonin receptor. The story here (human vs rat) might be a little complex, but worth including since SNPs in MTNR1B (encodes melatonin receptor 1B) have one of the strongest associations with fasting plasma glucose.

Section 2.1.10 Inclusion of Amisten et al, 2013, Pharmacology & Therapeutics, 139 (3) 359+ into the reference list would be helpful. This is one of the most comprehensive reviews of GPCRs expressed in pancreatic islets.

Section 2.2.3 The cytokine receptor section focuses on IL-1β and TNFα signalling, but you show IFNγ in figure 4 without exploring the signalling pathway in the text. This is somewhat of a glaring omission, IFN signalling (via STAT1) has significant importance in the regulation of beta-cell viability, with particular importance in T1D. A recent clinical trial has shown that JAK inhibitors are beneficial in T1D, and this is probably mediated partially through blockade of IFN signalling in the disease. As such, I would like to see a (short) section on IFN receptors.

Section 2.2.3 Following on from the point above. The section only seems to suggest that pro-inflammatory cytokines can stimulate beta-cells, and only mentions two of them. A better job needs to be done to get across that there are an array of different cytokines receptors on beta-cells. Whilst activation of some of these will be detrimental to these cells, other cytokines are protective (e.g. IL-13/4), and some impact beta-cell function (e.g. IL-6). I appreciate that it wouldn’t be sensible to write about every receptor, but you need to at least acknowledge some of these things.

Figure 2. The canonical JAKs involved in IFNγ signalling are JAK1 and JAK2; TYK2 is involved in type 1 IFN signalling. This is not properly reflected in the diagram.

A number of the figures have only a summary title as their legend. These need to be expanded, rather than rather than just stating ‘see text for more detailed explanation’. Many readers will want to use the figures independently of the manuscript text.

Comments on the Quality of English Language

Some review of the English language used in the review would be helpful. There are some small, but regular grammatical and language errors throughout. It’s important to state that these do not diminish the message, I felt that was very clearly delivered. However, I think clearing these up is important.
